# Historical and recent aufeis, the Indigirka river basin (Russia)

*Olga Makarieva[1,2,], Andrey Shikhov[3], Nataliia Nesterova[2,4], Andrey Ostashov[2]

[1]*Melnikov Permafrost Institute of RAS, Yakutsk*
[2]*St. Petersburg State University, St. Petersburg*
[3]*Perm State University, Perm*
[4]*State Hydrological institute, St. Petersburg*
*RUSSIA*
*omakarieva@gmail.com

**Abstract:** A detailed spatial geodatabase of aufeis (or naleds in Russian) within the Indigirka River watershed (305 000 km$^2$), Russia, was compiled from historical Russian publications (year 1958), topographic maps (years 1970–1980's), and Landsat images (year 2013-2017). Identification of aufeis by late-spring Landsat images was performed with a semi-automated approach according to Normalized Difference Snow Index (NDSI) and additional data. After this, a cross-reference index was set for each aufeis, to link and compare historical and satellite-based aufeis data sets.

The aufeis coverage varies from 0.26 to 1.15% in different sub-basins within the Indigirka River watershed. The digitized historical archive (Cadastre, 1958) contains the coordinates and characteristics of 896 aufeis with total area of 2064 km$^2$. The Landsat-based dataset included 1213 aufeis with a total area of 1287 km$^2$. Accordingly, the satellite-derived total aufeis area is 1.6 times less than the Cadastre (1958) dataset. However, more than 600 aufeis identified from Landsat images are missing in the Cadastre (1958) archive. It is therefore possible that the conditions for aufeis formation may have changed from the mid-20th century to the present.

Most present and historical aufeis are located in the elevation band of 1000 – 1200 m. About 60% of total aufeis area is represented by just 10% of the largest aufeis. Interannual variability of aufeis area for the period of 2001-2016 was assessed for the Bolshaya Momskaya aufeis and for a group of large aufeis (11 aufeis with a areas from 5 to 70 km$^2$) in the basin of the Syuryuktyakh River. The results of this analysis indicate a tendency towards an area decrease in the Bolshaya Momskaya aufeis in recent years, while no reduction in Syuryuktyakh River aufeis area was observed.

The combined digital database of the aufeis is available at https://doi.pangaea.de/10.1594/PANGAEA.891036.

**Keywords:** aufeis, Indigirka, Landsat, NDSI, Cadastre, Cadastral map, Bolshaya Momskaya aufeis

## 1. Introduction

Aufeis (naleds in Russian, icings in English) are accumulations of ice that are formed by freezing underground and surface waters on the surface of the earth or ice along streams and river valleys in arctic and subarctic regions. They affect water exchange and economic activity (Alekseev, 1987). Aufeis are found in permafrost regions such as Alaska (Slaughter, 1982), Siberia (Alekseev, 1987), Canada (Pollard, 2005), Greenland (Yde and Knudsen, 2005) and others (Yoshikawa et al., 2007). Aufeis formation can result in significant economic expenses as aufeis may negatively affect infrastructure and therefore natural resource extraction (Aufeis of Siberia…, Nauka, 1981). Moreover, the springs that often feed aufeis may in some cases be the

only source of water for remote communities (Simakov, Shilnikovskaya, 1958). In Russia, aufeis are found in the North-East, Transbaikal region, Yakutia, and West Siberia. Sokolov (1975) estimated that the total aufeis water storage in Russia to be at least 50 km$^3$, which approximately equals the Indigirka River total annual streamflow.

The main hydrological role of aufeis is the seasonal redistribution of the groundwater component of river runoff, where the winter groundwater discharge is released to summer streamflow through melting of aufeis (Surface water resources, 1972). In most cases, the share of the aufeis component in a river's annual streamflow accounts for 3-7%, reaching 25-30% in particular river basins with an extremely large proportion of aufeis (Reedyk et al., 1995; Kane & Slaughter, 1973; Sokolov, 1975). The most significant water inflow from aufeis melting takes place in May-June (Sokolov, 1975). For example, the share of the aufeis flow accounts for more than 11% of total annual streamflow at the Indigirka River (gauging station Yurty, 51 100 km$^2$). In May, aufeis melt may represent 50% of monthly total streamflow, but decreases in June to 35% (Sokolov, 1975).

It is important to understand how climate change may impact aufeis formation because warming has been observed in this region causing the transformation of permafrost (Romanovsky et al., 2007), glaciers reduction (Ananicheva, 2014) and hydrological regime changes (Bring et al., 2016; Makarieva et al., 2018). Aufeis are formed by a complex connection between river and groundwater. Many studies have reported the increase of minimum flow in Arctic rivers (Rennermalm and Wood, 2010; Tananaev et al., 2016), including those where aufeis are observed in abundance (Makarieva et al., 2018, in review). A widely accepted hypothesis for permafrost regions is that a warming climate increases the connection between surface- and groundwater that in turn leads to the increase of streamflow, both in cold seasons and in annual flow (Bense et al., 2012; Ge et al., 2011; Walvoord et al., 2012; Walvoord and Kurylyk, 2016). Variation and changes in aufeis extent can be assessed using remote sensing techniques, where aufeis dynamics can serve as an indicator of groundwater change that is otherwise difficult to observe (Topchiev, 2008; Yoshikawa et al., 2007).

The understanding of how aufeis respond to a warming climate varies. Alekseev (2016) suggests three to 11 year up and down cycles of aufeis maximum annual size, which may vary up to 25-30% in comparison with long-term average values. However, the same author (Alekseev, 2016) states a general tendency to the decrease of aufeis volume for the last 50-60 years in some aufeis-affected areas of Russia such as the Baikal region, South Yakutia, Kolyma region, Eastern Sayan Mountains, following the increase of global and local air temperature.

Some authors suggest that degradation of permafrost in the discontinuous and sporadic permafrost regions will lead to the decrease of the number of aufeis and even an almost complete disappearance. Meanwhile, in the zone of continuous permafrost in North-East Siberia, a climate warming of 2-3 °C is not projected to lead to significant changes in permafrost extent, but will increase the number and size of both through- and open taliks by the end of the 21th century (Pomortsev et al., 2010). Such a scenario may result in the reduction of area of large aufeis and formation of new small aufeis (Pomortsev et al., 2010).

In Alaska as well, no significant changes were documented in the area and volume of aufeis over the past few decades or even a century (Yoshikawa et. al, 2007). They suggested that the formation and melting of ice is less dependent on climate and more so on the source (spring) water properties such as temperature and volume.

In 1958, Simakov and Shilnikovskaya (1958) compiled and published a map inventory of aufeis of the North-East USSR (scale 1:2 000 000). Since then, there has been no update on the information on aufeis in this region, apart from some specific studies. In 1980-1982, an inventory of aufeis in the zone of the Baikal-Amur Mainline was published (Catalog of Aufeis…, 1980, 1981, 1982). Markov et al. (2017) summarized the results of field studies on aufeis in the southern mountain taiga of Eastern Siberia from 1976 to 1983. Grosse and Jones (2011) compiled the spatial geodatabase of frost mounds (or pingos) for northern Asia from topographic maps. Further, the glacier science community has mapped past and recent glacier

cover across the globe (GLIMS and NSIDC, 2005, updated 2017). However, as far as the authors are aware, no electronic catalogue of aufeis exists.

The aim of this study is to update the inventory of aufeis in the North-East of Russia using Landsat images, as well as to develop an electronic catalogue, which will contain data on historic and current location and characteristics of aufeis. Here we present work that has been completed for the Indigirka River basin (down to the Vorontsovo gauging station, 305 000 km$^2$).

The new database, which includes geographic information system (GIS) formatted files, is freely available (Makarieva et al., 2018) and can be used both for both scientific purposes and for solving practical problems such as engineering construction and water supply studies.

## 2. Study region

The study region is the Indigirka River basin, which is located in Northeastern Siberia and covers an area of 305 000 km$^2$ (Fig. 1). Most of the basin is represented by highlands with a number of mountain ranges (< 3 003 m) including the Cherskiy and Suntar-Khayata mountains. The lowland elevation reaches heights up to 350 m.

The climate of the study area is distinctly continental with annual average and lowest monthly air temperature varying from −16.1 and −47.1 ˚C, respectively, at the Oymyakon meteorological station (726 m, 1930-2012) to −13.1 and −33.8 ˚C, respectively, at the Vostochnaya station (1 288 m, 1942-2012). Most precipitation (over 60%) occurs in the summer season. Average annual precipitation at the Oymyakon weather station is 180 mm and at the Vostochnaya station 278 mm.

The Indigirka River basin is located in the zone of continuous permafrost. Permafrost depth can reach 450 m in the mountains, up to 180 m in river valleys and intermountain areas, with taliks found in river beds and fractured deposits. The hydrogeological regime is affected by the active layer, which varies from 0.3 m to over 2 m (Explanatory note …, 1991). The river runoff regime is characterized by high snowmelt freshet, summer-autumn rainfall floods, and low winter flow. In winter, small- and medium-sized rivers completely freeze. Freshet starts in May-June and lasts for approximately 1.5 months. Melt waters from aufeis, glaciers, and snow patches add to the river discharge in summer.

In total, about 10 000 aufeis with a total combined area of about 14 000 km$^2$ (Sokolov, 1975) are known in North-East Russia. The watershed area covered by aufeis varies from 0.4 to 1.3%, reaching 4% in some river basins (Tolstikhin, 1974). Most aufeis are of ground water origin; significantly less often they are formed out of river waters or are of a mixed type (Tolstikhin, 1974).

## 3. Materials and methods

### 3.1 The database of aufeis based on the Cadastre (1958) and topographic maps

The inventory map (scale 1:2 000 000) and the Cadastre of aufeis of the North-East of the USSR (Simakov, Shilnikovskaya, 1958), hereinafter referred to as the Cadastral Map and the Cadastre, became the first summarizing quantitative work on aufeis within the territory. The effort was carried out in the framework of the Central complex thematic expedition of the North-East Geological Survey of the USSR.

The Cadastre contains data on 7 448 aufeis of different size and over 2 000 boolgunyakhs (frost mounds). Of the total number of aufeis, 7 006 are plotted based on air-photo interpretation data, and another 442 on geological reports from field data. It should be noted that aufeis were identified based on geomorphologic features, meaning that in some cases only the areas or river valleys with aufeis were identified but not aufeis themselves.

In the Cadastre (1958) and our digitalization, the following characteristics of the aufeis are presented: location (the name of the river, the distance from the mouth or source), size (maximum length, average width, and area) and the dates of ice recording in aerial images (ranging from 08.06.1944 to 27.09.1945). Areas of the aufeis were evaluated via planimetering.

Only very large aufeis (> 3.3 km$^2$) were plotted on the Cadastral Map (1958), while the
others are shown as point locations. Each aufeis on the Cadastral Map (1958) has its
corresponding number, whose identifier and corresponding information can be found in the
Cadastre (1958). As noted by Simakov and Shilnikovskaya (1958), some very small aufeis
(<0.01 km$^2$) could have been missed due to their indecipherability on aerial images, or they
might have already melted by the time of the aerial photography. The example of the Cadastral
Map's sheet (1958) for the Indigirka River upper reaches is presented in fig. 2.
Here, we developed the GIS database of aufeis in the Indigirka River basin up to the cross-
section at the Vorontsovo gauging station based on the Cadastre (1958) and topographic maps.
Our compilation contains data on 896 aufeis. The aufeis are presented as point objects in our
database. The areas are specified for only 808 aufeis. The total area of all the aufeis with
specified area accounts for 2063.6 km$^2$ and the areas of individual aufeis vary from 0.01 to 82
km$^2$.
In the Cadastre, the dates of ice recording for 592 aufeis (66%) are presented, based on
aerial images within the study area. The average seasonal date of recording is August 2, ranging
from June 8 to September, 27. The dates of ice recording for the remaining 34% of the aufeis
were not described, meaning that aufeis detection could be carried out based not on the visible
ice presence at the aerial images but on geomorphological features of river valleys. Therefore,
the Cadastre might contain data on old aufeis glades, where the aufeis themselves were absent.
Spatial positioning of the Cadastral Map of aufeis was conducted using the location
description by Russian topographic maps with the scale of 1:200 000. Grosse and Jones (2011)
used the same set of maps for compiling the dataset of pingos (frost mounds) in northern Asia
and described those maps in details therein. The maps at 1:200 000 scale were based on more
detailed maps of 1:50 000 and 1:100 000 scale, which were derived from aerial photography
acquired in the 1970–1980's. The use of 1: 200 000 scale guarantees the position assessment
precision to within 100 m. Each map sheet was visually searched for aufeis and identified aufeis
were marked with an area polygon in a GIS layer. The locations of 330 aufeis (area 358 km$^2$)
were determined based on topographic maps. When digitized, a point was plotted in the middle
of an aufeis at a topographic map.
The locations of the remaining aufeis were determined with the positioned map of the
Cadastre. Additionally, 11 aufeis were found, which were absent in the Cadastre, but present in
the topographic maps. Aufeis areas were estimated by digitalization of the maps. Areas of the
remaining aufeis were estimated with the Cadastre. It was not possible to estimate the area of 88
aufeis, as they were not drawn on the topographic maps and only their location, but not area, was
stated in the Cadastre.
Table 1 contains the structure of the GIS dataset of aufeis according to the Cadastre.
**3.2 Identification of aufeis based on Landsat data**
Aufeis location and area are relatively easy to determine using Landsat and/or Sentinel-2
images, received immediately after snow cover melting. Snow and ice are known to be
characterized by relatively high reflectance in the visible and near infrared spectral bands and its
significant decrease in mid infrared band. Normalized Difference Snow Index (NDSI) is based
on this pattern and is calculated according to the formula (Hall et al., 1995):
$$NDSI = (GREEN - SWIR1) / (GREEN + SWIR1)$$,

where SWIR1 is reflectance in mid infra-red band (1.56 – 1.66 μm for the Landsat-8 images),
and GREEN is reflectance in the green band (0.525 – 0.6 μm for the Landsat-8 images).
Following Hall et al. (1995), the threshold value for snow and ice is set at 0.4. Apart from using
NDSI, other indices have been suggested to detect aufeis by Landsat images (but not used here).
These are Normalized Difference Glacier Index (NDGI) and Maximum Difference Ice Index
(MDII). Their advantages and disadvantages are discussed by Morse and Wolfe (2015).
Landsat-based detection of aufeis required some additional data to exclude other surface
types with similar spectral characteristics, such as snow-covered areas, turbid water, etc. It is

problematic to separate floodplain lakes from aufeis by late-spring satellite images, because many of these lakes are still ice-covered in May-June. Morse and Wolfe (2015) recommended creating a mask of water surface by mid-summer images (when all water bodies are already not covered by ice), to exclude them from further analysis.

Aufeis detection in the Indigirka River basin was carried out based on the Landsat-8 OLI satellite images, 2013-2017, downloaded from the United States Geological Survey web-service (https://earthexplorer.usgs.gov). We used Landsat 8 collection 1 level-one terrain-corrected product (L1T) with radiometric and geometric corrections. In total, 33 images completely covering the Indigirka river basin were processed. We selected late-spring images (between 15 May and 18 June), to detect the maximum possible number of aufeis, since in June they melt intensively. There was between 1-20% of cloudiness in some images.

Preprocessing of the images was performed with the use of Semi-Automatic Classification Plugin module (QGIS 2.18). It includes the calculation of surface reflectance and atmospheric correction by Dark Object Subtraction (DOS1) image-based algorithm, described by (Chavez, 1996).

The Aufeis detection algorithm was realized in ArcGIS with the help of the ModelBuilder application. Apart from the Landsat images, the digital terrain model (DTM) GMTED2010 (Danielson and Gesch, 2011) with a spatial resolution of 250 m was used to build a network of thalwegs within the study basin. This is essential for semi-automated separation of the aufeis from snow-covered areas in late-spring Landsat images. Indeed, almost all aufeis are located either at streams or thalwegs, or in immediate proximity to them. On the contrary, the snow cover in late spring mainly remains on mountain ridges and other elevated locations, i.e. relatively far from thalwegs. Based on the preliminary analysis of aufeis location in relation to the created network of thalwegs, we found that a 1.5 km wide buffer zone around the thalwegs covers almost all aufeis. So, snow and ice covered areas, which are located outside this buffer, are excluded from further analysis.

The process of aufeis detection by Landsat images consisted of the following steps:
- Detection of snow-ice bodies with the NDSI threshold of 0.4.
- Creation of a water mask with threshold values of the Normalized Difference Water Index (NDWI) (taken equal to 0.3), and reflectance in the near-infrared band (taken equal to 0.04).
- Extraction of the detected snow-ice bodies by the buffer zone around thalwegs (1.5 km wide).
- Conversion to vector format, area calculation and removal of objects smaller than 5 Landsat pixels (0.45 ha).

The suggested algorithm allows successful aufeis detection if an image is predominantly snow-free. At the end of May/early June, many aufeis in mountain regions are still covered by snow. Their detection required later images, obtained in mid-June.

Morse and Wolfe (2015) suggested a new spectral index MDII for automatically distinguishing snow bodies from ice ones. However, here some of the high elevation aufeis were partially covered with snow at the image acquisition time. Instead of automatic processing, the outlining of high elevation aufeis was conducted manually when snow cover was present, with separation of aufeis from adjacent snow covered areas.

Further, during melt season, the aufeis often divide into several neighboring areas. When assessing the number of aufeis with satellite data, it is therefore necessary to aggregate the areas into one aufeis, if they are located at a distance <150 m (or five Landsat pixels) from each other, and within one aufeis glade.

As a result of semi-automated processing of Landsat images, aufeis with a total area of 1 253.9 km$^2$ were detected. During the subsequent comparison with the Cadastre data (see section 3.3 for more details), over 100 aufeis, with a total area of 33.5 km$^2$, were delineated manually. The gaps were mainly due to the presence of snow cover and/or cloud coverage in the images. To reduce the number of gaps, two to three images of the same territory were used. The total

number of aufeis, identified with the Landsat images in the Indigirka River basin, was 1 213 and their total area 1 287.4 km$^2$. Therefore, an omission error of automatic aufeis detection can be estimated as 2.7% of their total area.

The structure of the GIS dataset of aufeis according to Landsat images is presented in Table 2.

### 3.3.Cross reference between historical and satellite-based aufeis data collection

Cross-verification of aufeis data collections by the Cadastre (1958) and satellite imagery was performed in two steps. At the first step, we found the closest aufeis in the Landsat-derived dataset for each aufeis from the Cadastre data if the distance between them was less than 5000 m. The determination of search radius was based on a preliminary analysis of the aufeis locations by the Cadastre in relation to Landsat-based dataset. As a result, the cross index (identifier of the closest aufeis in the Landsat-derived dataset) and minimum distance (m) to the closest aufeis were determined for aufeis from the Cadastre. For Landsat-based dataset, the cross index is the key field for the reference to the dataset from the Cadastre.

At the second step, a full manual verification was performed to find the mistakenly interrelated aufeis. For example, if the closest aufeis from the Cadastre and from the Landsat-based dataset were at a distance of less than 5000 m, but in different thalwegs, they were considered as different (unrelated) aufeis.

In total, 260 aufeis from the Cadastre were not verified by Landsat images. For them, the NoData value (–9999) was set in the Cross Index and Distance fields of attributive table (see Table 1 with the structure of GIS dataset from Cadastre).

### 4. Results

### 4.1 Comparison of the historical and modern data collection

The results of the comparison are presented in Table 3. In total, 634 aufeis from the Cadastre were found by the Landsat images. They correspond to 611 aufeis identified with the images, meaning that in 23 cases, one aufeis in an image corresponds to two aufeis in the Cadastre. But 262 aufeis from the Cadastre were not detected by the satellite images. Those are mainly small aufeis, which melt by the middle of June. However, among them there are also 43 large aufeis over 1 km$^2$ (fig. 3-a). It is likely that since the mid-20th century, when the field observations were conducted and the Cadastre of aufeis was compiled, some aufeis could have disappeared.

A little over half of the aufeis detected by Landsat images are included in the Cadastre: a total of 602 aufeis detected (the total area of 250.4 km$^2$) are not included in the Cadastre (fig. 3-b). Such a significant difference can be caused by the following reasons:

1. In some cases a single aufeis, according to the Cadastre, corresponds with two or more aufeis by satellite image;

2. Aufeis are characterized by significant interannual variability, which results in possible formation of new aufeis in areas where they previously were not observed (Alekseev, 2015; Pomortsev et al., 2010; Atlas of snow…, 1997).

Total aufeis area evaluated based on satellite images, appeared to be 1.6 times smaller than stated in the Cadastre (1958). First and foremost, such difference can be explained by the fact that it was not the area of the aufeis themselves, but instead the aufeis glades, that were reported in the Cadastre (1958) and this corresponds to the maximum aufeis area during one or several seasons. With the satellite data, the areas of the aufeis themselves were assessed and when mid-June images were used, the aufeis area was significantly smaller than the typical annual maximum.

Aufeis area distribution according to the Cadaster and satellite data is shown as Lorenz curves (fig. 4). In both cases, the shape of the curves signifies a high degree of irregularity which

is similar: 10% of the largest aufeis make up 61 and 57% of their total area according to the
Landsat and the Cadastre data, respectively.
The cross-verification of the Cadastre and satellite data show that almost 60% of aufeis
that are unconfirmed in the Landsat imagery and that are therefore only present in the Cadastre,
have an individual aufeis area less than 0.25 km$^2$ (Fig. 5-a). The confirmed aufeis account for
about 20% of the area stated in the Cadastre. Thus, it was mainly small aufeis that were not
confirmed in the Landsat images. Conversely, Fig. 5-b shows that almost 60% of the aufeis
detected in the Landsat images but not listed in the Cadastre have an area each of less than 0.25
km$^2$.

## 311 4.2. Aufeis distribution by elevation

In general, aufeis distributions by elevation as assessed with the Cadastre and Landsat data
are quite similar, although there are some differences that are elevation-specific (fig. 6). Most
aufeis are located in the elevation band of 1 000 – 1 200 m. At lower elevations (up to 800 m)
the number of aufeis according to Landsat data is higher than stated in the Cadastre. At the
elevations of 1 400-2 000 m, more aufeis are identified in the Cadastre data than by the satellite
images. This can be explained by the fact that many aufeis located at high altitudes often have a
small area, so they could have been missed during the analysis of the satellite data. Further, they
could have been covered with snow at the image acquisition time, which would increase the
possibility of them being missed.
The elevation band of 200-300 m is characterized by the location of large aufeis. Though
less than 2.5% and 5.0% of aufeis by the Cadastre and Landsat images are situated here, they
represent about 11 and 13% of aufeis area from the datasets respectively (fig. 7).

## 325 4.3 Aufeis distribution by river basins

In the Indigirka River basin, there are several zones with a high density of aufeis: in the
southern part (the Suntar and Kuidusun Rivers basins), as well as in the central part (Chersky
Range slopes) (fig. 8). The largest aufeis identified with satellite images are located in the
Syuryuktyakh River basin on the north-east slopes of the Chersky Range. Meanwhile, aufeis are
almost absent in the northernmost (lowland) part of the Indigirka basin.
We analyzed the aufeis coverage for six river basins with available streamflow data. The
headwater part of the Indigirka River, with the gauge near the Yurty village (area 51 100 km$^2$), is
the basin with the largest aufeis coverage (Table 4). Correlation between average elevation of the
basins and their aufeis coverage (expressed as a percentage) is statistically significant. Among 6
basins, the Spearman rank correlation coefficients between the basin average elevation and
aufeis percentage are 0.71 and 0.77 by the Cadastre and satellite data, respectively.

## 337 4.4 Aufeis area interannual variability

The assessment of aufeis area interannual variability was conducted in two areas: for the
Bolshaya Momskaya aufeis, which is located in the Moma River channel (area in the Cadastre is
82 km$^2$), and for a group of large aufeis (total area in the Cadastre is 287.8 km$^2$) in the
Syuryuktyakh River basin, which is the left-bank tributary of the Indigirka River.
Cloudless images from Landsat-5 (TM), Landsat 7 (ETM+) and Landsat-8 (OLI) were
used with the acquisition dates between May 1 and June 30. In the USGS archives, there are no
Landsat-5 images for the study territory for the 1984-2007 period. This limits the duration of
satellite observations on aufeis to the period since 1999 (when the Landsat-7 satellite was
launched). Also, the clouds complicate the acquisition of representative data. The list of the
acquisition dates and assessed aufeis area values are presented in Table. 5.
Both areas are located at low elevations (Bolshaya Momskaya 430 to 500 m and
Syuryuktyakh 200 to 500 m), which contributes to the relatively early and intensive aufeis melt
in spring. The aufeis reach their maximum area by the beginning of May. Using the available
satellite images it is impossible to make a reliable conclusion on aufeis area increase or decline,
because the acquisition dates vary significantly from year to year. However, it is possible to
make some conclusions based on the available data:
1. In 2002-2017 the Bolshaya Momskaya aufeis did not reach the maximum area stated in
the Cadastre ($82 \text{ km}^2$), even though the satellite image was acquired during the first week of May
(2005) when aufeis melting had not yet started. Comparing two images, taken in similar
conditions (08.05.2005 and 15.05.2013), it was found that aufeis area in 2013 was smaller by
$18.1 \text{ km}^2$ than in 2005. Accordingly, the Bolshaya Momskaya aufeis may have seen a decreasing
trend over time in its maximum coverage.
2. The area of the largest aufeis in the Syuryuktyakh River basin in May 2014 was 78.0
$\text{km}^2$, which is $8 \text{ km}^2$ larger than stated in the Cadastre. One may note also that the maximum
aufeis areas in the Syuryuktyakh River basin were detected by the images received at the end of
the period (2014-2017), including mid-June (18.06.2015). Therefore, it can be suggested that the
aufeis areas within Syuryuktyakh River basin have not decreased since 2002.
**5. Discussion**
The most important uncertainty in the obtained results relates to our ability to draw a
conclusion on the long-term trend of total aufeis area comparing the historical and satellite-
derived datasets. The total area of aufeis estimated by Landsat images is 38% less than according
to the Cadastre. Is it possible to confirm that such a significant reduction in the aufeis area really
occurred? Considering this issue, it is important to emphasize some limitations of the
methodology and the created datasets.
The main limitation of the historical aufeis dataset is that the Cadastre provides an area of
aufeis glades, but not the aufeis themselves. Simakov and Shilnikovskaya (1958) noted that the
areas of aufeis glades match the average annual maximum of the ice-covered area. Alekseev
(2005) states that the assessment of the stages and patterns of the development of aufeis glades
based on the analysis of their landscape and geomorphological features is difficult due to the lack
of research on temporal aspects of mutual transitions of landscape facies and their factorial
dependencies. However, studying the aufeis landscapes in the central part of Eastern Sayan
Mountains, Alekseev (2005) assumes that the vegetation community which is a typical indicator
of aufeis development may persist for 200-300 years after the beginning of aufeis processes
attenuation.
The satellite-derived assessment of the aufeis area has the following main source of
uncertainty. It is often impossible to determine the maximum area of aufeis by satellite images,
since it is observed at the beginning of snow melt season, when aufeis are still covered with
snow. In late spring and beginning of summer, the area of aufeis may already been significantly
reduced in comparison with the maximum values, due to melting and mechanical destruction.
Maximum intensity of aufeis melt in the studied region is observed in June when spring
flood river streams actively erode the aufeis surface. Sokolov (1975) reported the results of the
observations at the Anmyngynda aufeis carried out in 1962-1965. This aufeis is located in the
upstream area of the Kolyma river basin (723 m a.s.l.) and may be used as being representative
of the mountainous part of the studied region. In 1962-1965, the aufeis area changed from 5.1 to
$6.2 \text{ km}^2$ with mean maximum area of $5.7 \text{ km}^2$. Aufeis melt has been observed to begin on
average on the $10^{\text{th}}$ of May. During May, the aufeis area decreased by 15% of the total area on
average. At the end of June, the remaining area was 34% of the maximum, i.e. during this month
more than 50% of the aufeis area has been destroyed. In the period from July to September, the
melting slowed down: in July the aufeis decreased by 22%, in August by 8%, in September by
3%. The area of aufeis at lower absolute elevations decreases faster at first half of the summer,
and in the upstream areas – in the second one (Sokolov, 1975).

Some aufeis in the mountainous regions could be missed by satellite images, since they
can be covered with snow until the end of June. However, their contribution to the total area is
non-significant.

Taking into account all the above-described limitations, and also that more than 600
aufeis that were missing in the Cadastre were found by Landsat images, we conclude that it is
not correct to make a conclusion about long-term trends of aufeis area based on the entire created
dataset. Following Pavelsky and Zarnetske (2017), we decided to examine only several of the
largest aufeis deposits in order to identify the long-term trend.

We selected the 38 largest aufeis with an area $\geq 10$ km$^2$ according to the Cadastre dataset,
confirmed by satellite data. Their total area decreased from 858.1 km$^2$ according to the Cadastre
to 356.3 km$^2$ according to recent Landsat images. Conversely, we also selected the largest aufeis
according to satellite data (18 aufeis with satellite-estimated area $\geq 10$ km$^2$). Their total area also
decreased significantly (from 428.6 km$^2$ according to the Cadastre to 343.5 km$^2$ according to
Landsat images). We also analyzed 8 giant aufeis with areas $\geq 35$ km$^2$ according to the Cadastre
dataset. They all were confirmed by the satellite images; however seven of the eight had a
significantly smaller area (from 2 to 21 km$^2$) with the decrease being 2-10 times. Only one giant
aufeis in the Syuryuktyakh River basin has the area by Landsat larger than by Cadastre, at 72 and
64 km$^2$ accordingly. It should be noted that the formation of new (mainly small) aufeis can
slightly reduce the rate of the aufeis area decrease.
**6. Conclusion**
The research conducted here is the first step of the study aimed at the development of a
GIS database of the aufeis of North-East Russia. Historical data of the Cadastre (1958) and
topographic maps were used to create a geodatabase of aufeis in the Indigirka River basin (up to
the Vorontsovo gauge, with the area of 305 000 km$^2$). It contains historical data on 896 aufeis
with total area of 2063.6 km$^2$. Aufeis detection was conducted for the 2013-2017 period using
Landsat imagery with 1213 aufeis identified having a total area of 1287.4 km$^2$. The historical
dataset from the Cadastre (1958) and more recent satellite-based dataset were compared and
combined in the joint Catalogue of aufeis within the Indigirka River basin, available at the
PANGAEA repository (https://doi.pangaea.de/10.1594/PANGAEA.891036).
Recent total aufeis area is 1.6 times smaller than stated in the Cadastre (1958). The more
significant changes occurred to 38 large and giant aufeis (area $\geq 10$ km$^2$) with total decrease of
area by 501.8 km$^2$ (or 66% of the total reduction). Simultaneously, the historical Cadastre
archive is lacking data on over 600 aufeis that were identified using satellite images. This
suggests that the Cadastre data is incomplete, while there may also have been significant change
in aufeis formation conditions in the last half century.

The analysis of large and giant aufeis seems to indicate that there has been a significant
decrease in aufeis area over the period of last 70 years. Additional analysis of historical aerial
photography data could help to clarify the issue of aufeis area decline trend since the middle of
the 20th century to the present. One of the further study goals will be to find out the extent to
which these changes are climate-derived and to identify their impact on river streamflow.

**Acknowledgements.** The authors are grateful to David Post, Anna Liljedahl and an
anonymous reviewer for valuable comments and assistance with English.

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

Table 1 The structure of GIS database of aufeis by Cadastre (1958)

| Field name | Field alias | Description |
|---|---|---|
| FID | FID | Index number (Object ID) |
| AufDataSrc | Aufeis data source | Aufeis Cadastre data (1958) (for all objects) |
| Auf_area | Aufeis area Cadastre (km$^2$) | Aufeis area (km$^2$) from the Cadastre (1958). If the data was missing, the area was calculated by topographic maps (1980) scale 1: 200 000 |
| Auf_index | Aufeis index Cadastre | Index of the aufeis in the Cadastre (1958) (it contains 0 if the aufeis was missing in the Cadastre, but found in the topographic map (1980) scale 1: 200 000) |
| Map_index | Cadastre map index | Index of the Cadastre (1958) map |
| Auf_topo | Aufeis in topo | Presence of the aufeis at topographic map (0 – missing, 1 – present) |
| Auf_in_map | Aufeis in map | Presence of the aufeis in the Cadastre (0 – missing, 1 – present) |
| Toponumber | Topo number | Nomenclature of the topographic map sheet |
| Date | Date | Date of fixing the presence of ice within the aufeis |
| Long | Long | Longitude, degree |
| Lat | Lat | Latitude, degree |
| Elevation | Elevation | Height above sea level (determined by Aster GDEM), m |
| Comment | Comment | Comments (mainly typos in the Cadastre map, or the method of determining aufeis area) |
| CrossIndex | Cross index | Cross index of aufeis derived from Landsat (if aufeis is not in Landsat, the value is missing) |
| Distance_m | Distance (m) | Minimum distance between the aufeis from the Cadastre and the same aufeis from Landsat image (m) |


Table 2 The structure of GIS database of aufeis by Landsat images (2013-2017)

| Field name | Field alias | Description |
|---|---|---|
| FID | FID | Index number (Object ID) |
| AufDataSrc | Aufeis data source | Landsat images (for all objects) |
| WRS2_ID | Landsat WRS2_ID | The Landsat scene identifier in the WRS2 graph of the US Geological Survey (USGS). The first three digits indicate the column number, and last three digits represent the line number. |
| Image_Date | Landsat image date | The date of image |
| Comment | Comment | Additional information, for example, if the aufeis was partly covered by clouds and additional images were used to estimate the area |
| CrossIndex | Cross index | Identifier of aufeis by Landsat images (key field for the reference to the Cadastre data) |
| Auf_Area | Aufeis area (km$^2$) | Aufeis area by Landsat image, km$^2$ |
| Elevation | Average elevation | Average elevation of aufeis, calculated by Aster GDEM digital elevation model |


Table 3 Data correlation of aufeis based on the Cadastre (1958) and the Landsat images

| Data source | Matching aufeis number and area (km$^2$) | Not confirmed aufeis number and area (km$^2$) |
|---|---|---|
| Cadastre (1958) | 634 (1905.0) | 262 (158.6) |
| Landsat | 611 (1037.0) | 602 (250.4) |


Table 4 Aufeis area coverage (percentage) in the sub-basins within the Indigirka River watershed
by the Cadastre and Landsat data

| River | Area, km$^2$ | Average elevation, m a.s.l. | % aufeis coverage (Cadastre) | % aufeis coverage (Landsat) |
|---|---|---|---|---|
| Suntar River –Sakharinya River mouth | 7680 | 1460 | 0.97 | 0.78 |
| Elgi  – 5 km upstream of the Artyk-Yuryakh River mouth | 17600 | 1104 | 0.49 | 0.23 |
| Nera – Ala-Chubuk | 22300 | 1174 | 0.32 | 0.26 |
| Indigirka – Yurty | 51100 | 1256 | 1.15 | 0.80 |
| Indigirka – Indigirskiy | 83500 | 1185 | 0.82 | 0.56 |
| Indigirka – Vorontsovo | 305000 | 803 | 0.68 | 0.41 |


Table 5 Aufeis area changes, 2001-2017.

| Bolshaya Momskaya aufeis | | The group of aufeis in the Syuryuktyakh River basin | |
|---|---|---|---|
| Imagery date | Aufeis area, km$^2$ | Imagery date | Aufeis area, km$^2$ |
| 17.06.2002 | 29.2 | 26.06.2001 | 69.7 |
| 08.05.2005 | 66.2 | 29.06.2002 | 100.6 |
| 27.05.2006 | 57.9 | 04.06.2007 | 155.1 |
| 19.06.2009 | 39.5 | 17.06.2009 | 117.5 |
| 25.05.2011 | 61.7 | 22.06.2011 | 89.5 |
| 27.05.2012 | 49.6 | 21.05.2014 | 268 |
| 15.05.2013 | 48.1 | 18.06.2015 | 164.8 |
| 18.06.2017 | 21.9 | 04.06.2016 | 206.4 |


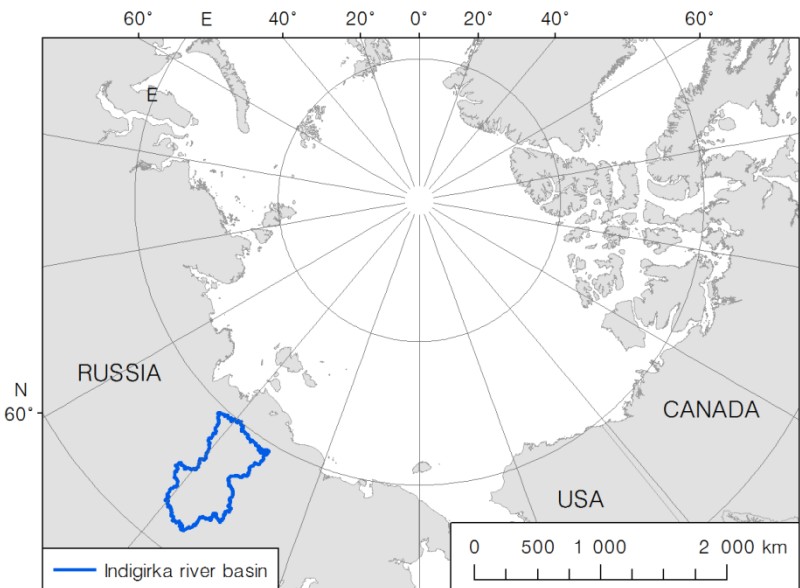

Fig. 1 Geographical location of the Indigirka river basin

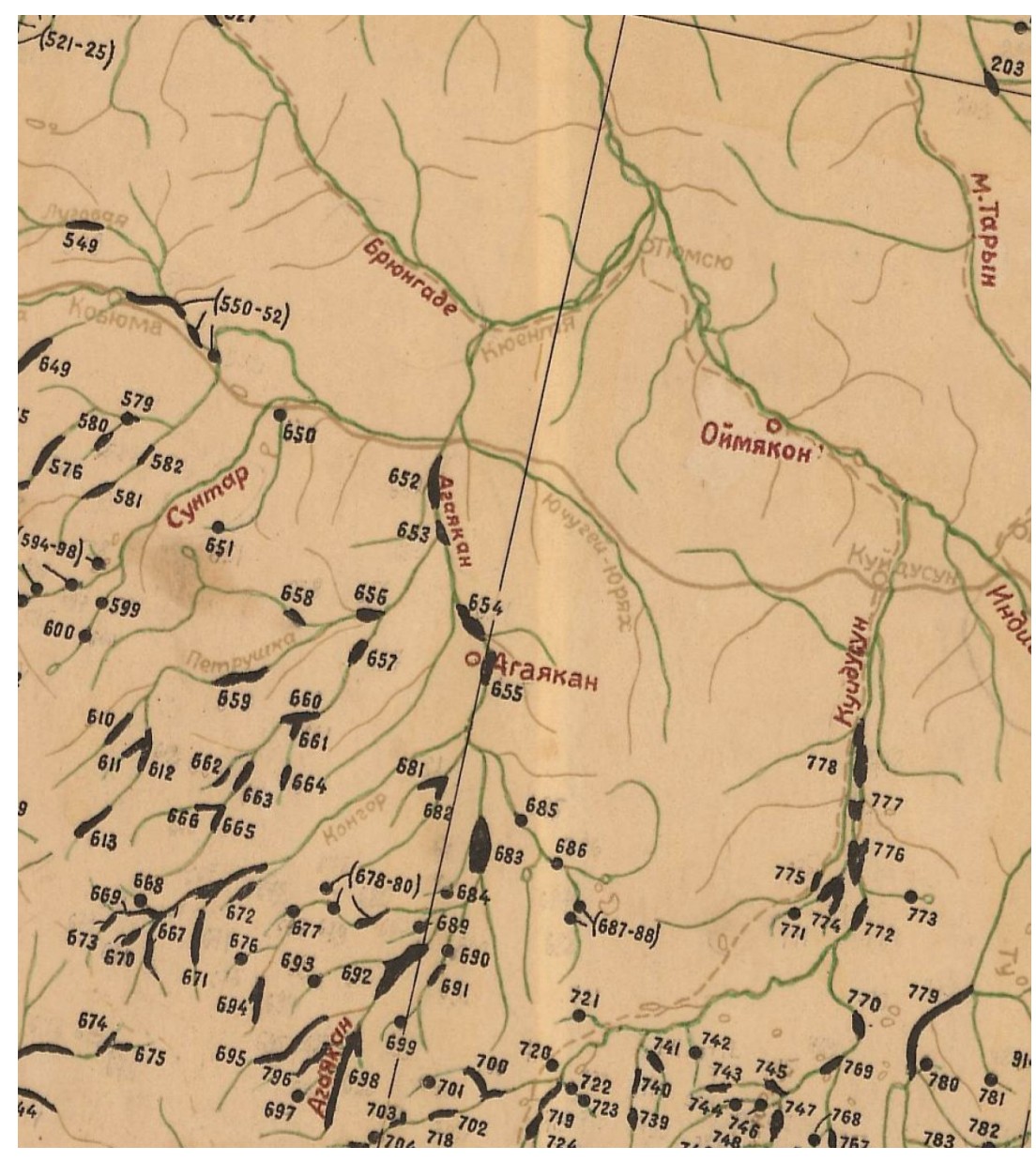


Fig. 2 Subset of the Cadastral Map of the North-East of the USSR from 1958 (sheet 7, upper reaches of
the Indigirka River – the basins of the rivers Suntar, Agayakan and Kuydusun).

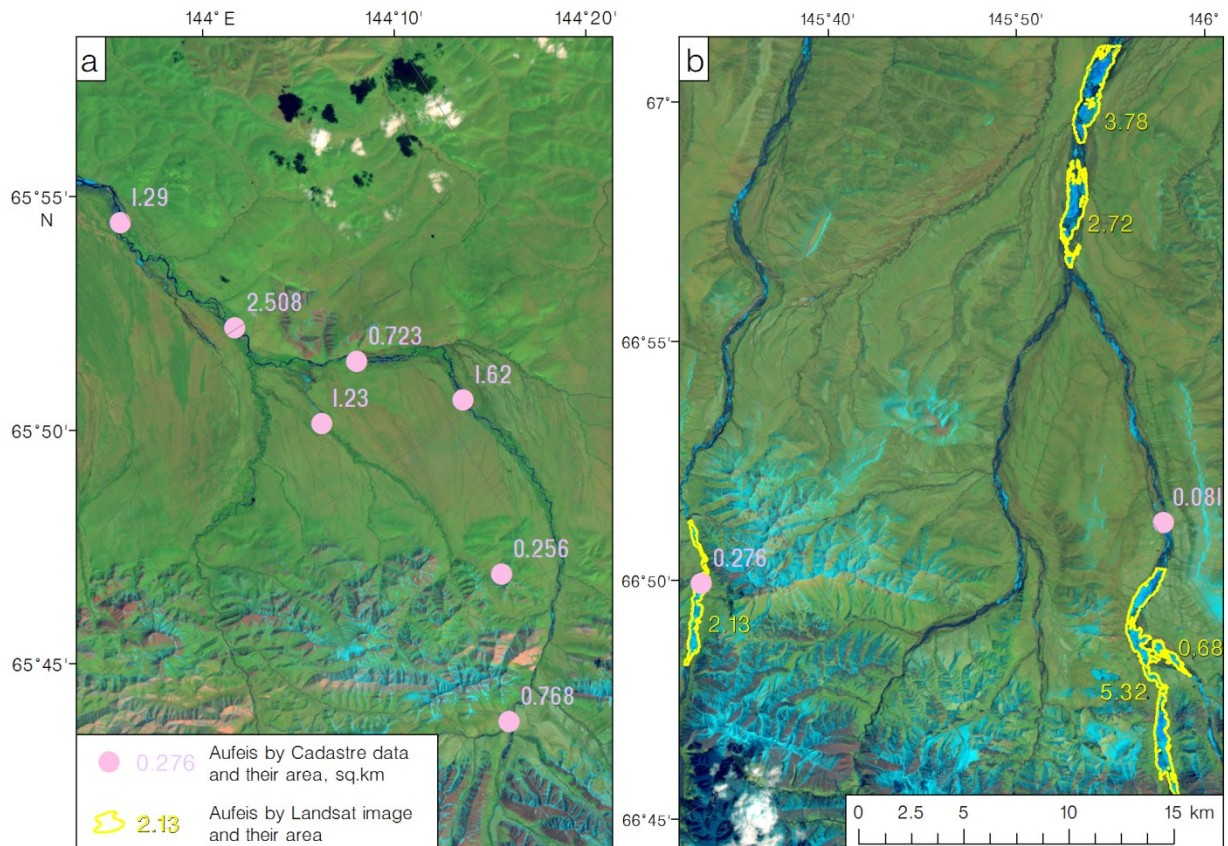


Fig. 3 Difference between aufeis location according to the Cadastre and satellite data: a) – aufeis are
absent in the image but present in the Cadastre (Landsat-8 image of 18.06.2017); b) – aufeis are absent
(or their area is understated) in the Cadastre but present in the image (Landsat-8 image of 30.05.2016).


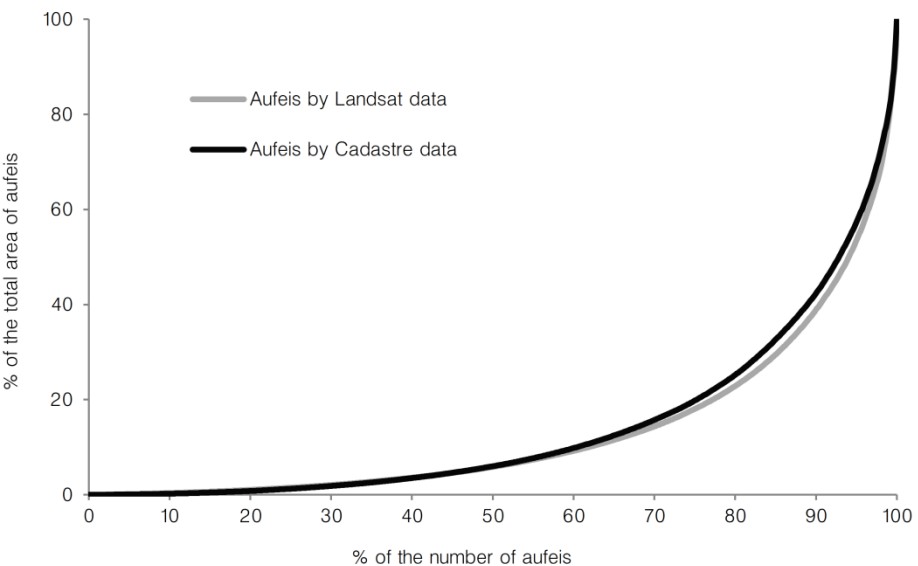


Fig. 4 Lorenz curves illustrating aufeis area distribution according to the Cadastre and Landsat data

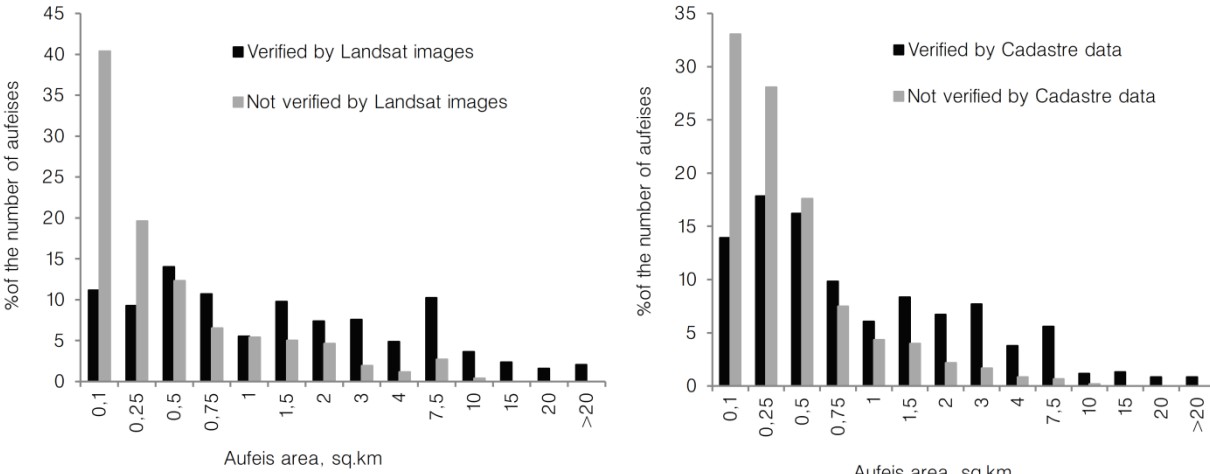


Fig. 5 Aufeis area distribution: a) – according to the Cadastre data, confirmed and not confirmed by
Landsat images, b) – according to Landsat images, confirmed and not confirmed by the Cadastre.



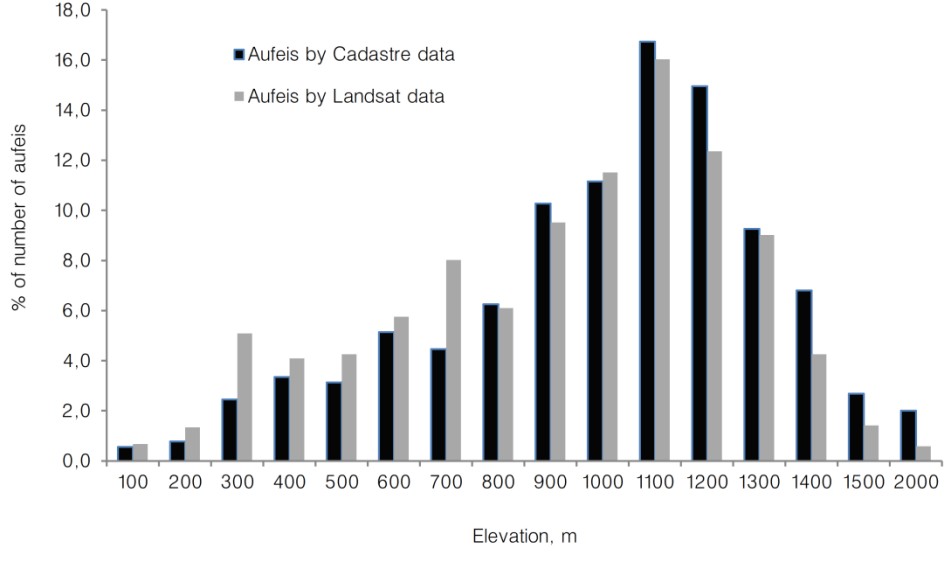


Fig. 6 Aufeis distribution by elevation within the Indigirka River basin.

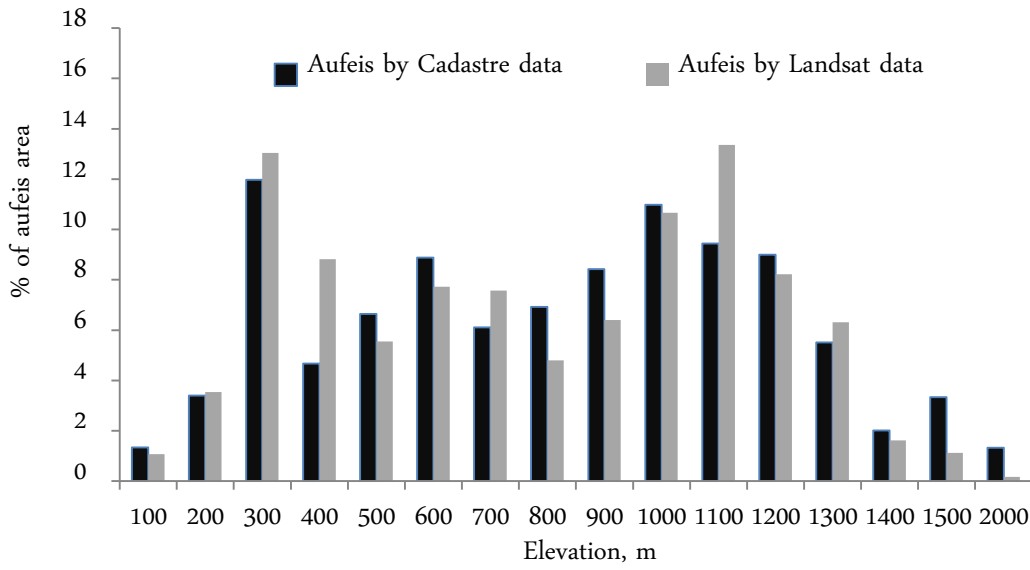


Fig. 7 Aufeis area distribution by elevation within the Indigirka River basin.


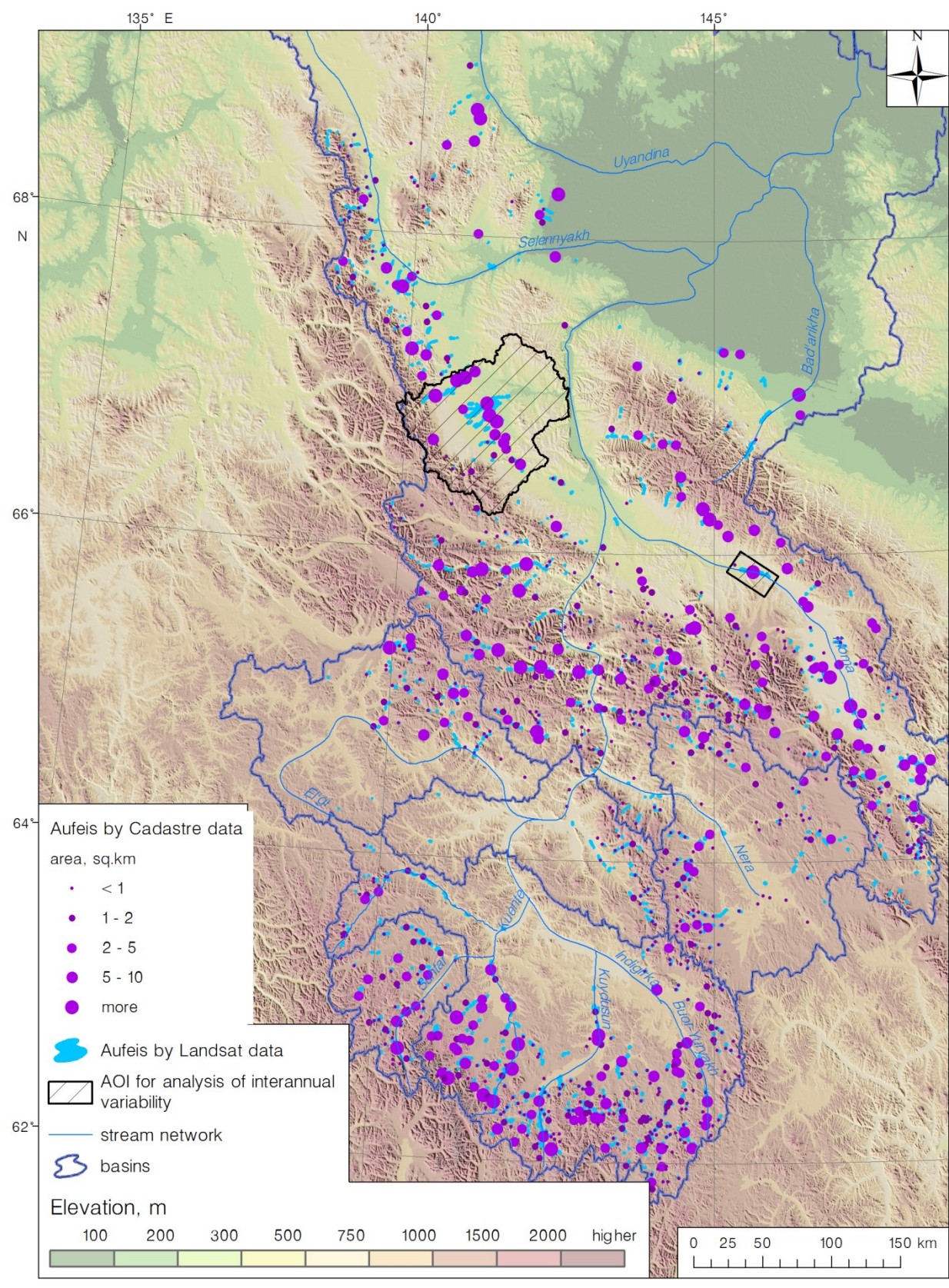


Fig. 8 Aufeis in the Indigirka River basin according to the Cadastre and Landsat images. Black
outlines with section lining represent the zones where aufeis area interannual variability was
assessed