# Peer review of "Historical and recent aufeis, the Indigirka river basin (Russia)"

_Earth System Science Data, 2018_

## Short Comment (SC1) · 21 Sep 2018

This is a fascinating analysis and I congratulate the authors on an excellent paper. I am very interested in where they might take this analysis from here. It seems to me that based on their data, there appears to be a declining trend in the number and/or size of aufises in the study area. I am wondering if the authors could access additional satellite and/or aerial photography or other historical records that might allow them to determine if such trends are in fact occurring and attempting to quantify them.

---

## Referee Comment (RC1) · Anonymous Referee #1 · 9 Dec 2018

This is a unique temporal data collection of aufeis data in the Indigirka river basin, Russia. Aufeis or naleds deposits are thick accumulations of ice that form during winter along stream and river valleys in arctic and subarctic regions impacting hydrology and geomorphology of these regions. The authors compiled and standardized historical data on aufeis deposits in the Eastern Siberian Indigirka river basin from a historical Russian National cadastre complementing data using historical topographical maps and added a new data set on aufeis derived from Normalized Differential Snow Index (NDSI) index calculation using Landsat 8 OLI sensor data. The authors cross-referenced the historical and the present-day data collection. The data collection is organised as a Geographic Information System GIS data base including data on lo-

cation, area coverage, elevation, time stamp, source of data in form of attribute tables and the aufeis objects in the data format of GIS point and polygonal vector layers. The Indigirka aufeis catalogue is published on PANGAEA in the form of a GIS data base with a helpful and detailed read-me description of the attribute tables. The data collection will be of interest to hydrologists, climatologists, geomorphologists, cryologists and social science. The authors document in the manuscript the generation of the historical and the modern date data sets and its meta data characteristics. The authors also discuss the validity of data, the cross referencing between historical and nowadays aufeis deposits and reasons for mismatches in areal coverage and locations and possible changes due to climate. The paper is in general clearly written with many details provided. However, the article including the title, the PANGAEA data publication including title, abstract and the metadata description need to be carefully edited for English before acceptance of the paper. The data compilation process and metadata is not thoroughly and clear enough shown and explained and the GIS data require further standardisation and optimization to make them reusable.

Technical issues, GIS data: 1) the GIS shape files contain different projections: The GIS data catalogue is published in PANGAEA as an ARCGIS project data base. The downloaded data base is userfriendly readible and usable using the proprietary GIS software ARCGIS. ARCGIS licences are costly and many user groups may use open source GIS or other geodata software packages. Using ARCGIS software the shapefiles are automatically but only virtually brought to the same projection. The GIS shape files are also readable and reusable using open source geodata software packages – however the 2 data collections have different projections (the aufeis kadastr shape file contains the projection "Asia_North_Lambert_Conformal_Conic" the aufeis Landsat shape file not). This requires users of these datasets who are using free software packages to reproject the shape files to a common projection prior to being able to use the data sets together. Please standardise the shapefiles using one projection

2) the GIS attribute files do not contain self-explanatory attribute names: The Indigirka

aufeis data collection is a highly valuable data set, specifically also because the authors are using cross reference indices to link the data sets. This needs to be made more clear in the naming and cross-referencing of the attribute names. E.g., the cross reference index should be also named accordingly, e.g. as cross index similarly in both attribute tables, not named ID in the aufeis_Landsat data set and named PolygonID in the aufeis kadastr data set. Naming of similar attributes should be standardized between the data sets, e.g. the attribute area in sqkm. Suggestions on attribute naming is attached as supplement. Please consider to change attribute names to more self-explanatory names. The data set can also be uploaded in Google Earth with visualisation of the data objects and the metadata and will be by this very easy re-usable if attribute naming and cross-referencing between the 2 data sets will be made as self-explanatory as possible.

3) consistency of published GIS data with manuscript content: Authors show in the manuscript assessments of both data sets – cadastre derived and satellite derived related to elevation. The attribute elevation is however missing in the attribute table of aufeis_Landsat. Consider to add information on elevation into the attribute table of the aufeis Landsat data set.

—- Issues, data publication on PANGAEA: Title: aufeis is the plural form of aufeis, the plural form aufeises does not exist. Abstract: The abstract should be extended to contain more technical information on the data. Authors should inform the users that the data download will consist of a complete ARCGIS project containing 2 different feature GIS shape files with historical and the nowadays aufeis data collection. The authors can add short information in the abstract on how the data were generated. Very useful for future users of the GIS data is to provide in the abstract text information on the projection of the GIS data collection – this is sometimes handy for reading data in in some open source geodata software packages. The authors could add an overview figure of the data set as additional information. Published data: the authors published the GIS project with 2 feature layer data and the 2 data collections also in

form of ASCII files and a detailed read me word file documenting the attribute tables. Information on the GIS project itself in the read-me file is missing: e.g., information on the format (ARCGIS) and projection.

—- Issues, manuscript:

General: aufeis is the plural form of aufeis, the plural form aufeises does not exist. Authors could also consider to sometimes refer to aufeis deposits in the manuscript if this fits. Authors could refer to the cadastral map instead of map throughout the text, also to better distinguish for the reader the cadastral map from topographic map forms.

Abstract: The authors should enrich the abstract with much more information on the technical generation and technical contents of the data set and with less discussion on changed areas and potential reasons that would be kind out of scope and not the focus of this ESSD publication. A great meta data information in this data collection is the cross-reference index enabling users of this data set to link and compare these very different 2 data set types: the historical and the nowadays aufeis data sets.

keywords: reconsider the keywords, e.g., aufeis, Indigirka, Bolshaya Momskaya, Landsat, NDSI, cadastre, cadastral map;

1 Introduction: authors should provide an explanation what is aufeis in the first sentences of the introduction. That aufeis are thick accumulations of ice that form during winter along stream and river valleys in arctic and subarctic regions.

2 Research objective: this subtitle is misleading as the motivation of this study and data set compilation is already well introduced by the authors in the introduction chapter. This chapter describes the study region. Please add an overview figure with the geographical setting of the Indigirka river basin and the extent of the data set in relation to Eastern Siberia. E.g., Figure 6 is already to zoomed in to provide this information.

3 Material and Methods: The authors should add the tables from the published read me file in the respective subsections 3.1 and 3.2. The authors should add flow chars

to make their data processing steps more clear in the in the respective subsections 3.1 and 3.2. For example the role of the thalweg creation remains unclear to the reader. The ASTER GDEM data set needs to be introduced and explained as the meta data information on elevation is taken from this digital data set. Also for the Landsat derived dataset? This does not become clear to the reader. 3.2. The level of the USGS Landsat data product that was used remains unclear. The authors did not use the Landsat T1 Level2 (L2) that is the surface reflection coefficient already? Did the authors use the Landsat T1 Level1 data products that are terrain-corrected (T1) and Top-of-Atmosphere radiances (L1)? Because authors refer to brightness? The authors describe: Preprocessing of the images (transformation brightness into reflection coefficient) was performed with the use of Semi-Automatic Classification Plugin module in QGIS 2.18. Does it mean that an atmospheric correction was performed to surface reflection coefficient? Which type of atmospheric correction was performed to come to the surface reflection coefficient / surface reflectance? 3.3 A good description of the cross reference between the aufeis deposits in the historical aufeis data collection and the nowadays data collection is missing. Authors can consider to add a short sub-paragraph 3.3. It would be helpful for re-using the data set if authors put some details here, e.g. highlight that there is the cross reference ID in both attribute tables.

4 Results and verification: The chapter does not seem to describe or focus on verification? In the first section of 4 Results the authors very interestingly assess the linkages and differences between the data sets – this could become a subchapter 4.1. with a title relating to the comparison of the historical to the morden data collection. All of the above points can be addressed with minor corrections, just a few sentences or less.

consider adding a Discussion chapter with a short discussion about the usability of this data set on aufeis area growth or decline, could be one outcome of your study on the variability to assign higher variability and lower accuracy to the extraction of the aufeis area at lower elevation? Would it be possible to assign different reliability (consistency of measurement) levels for the representativeness of the derived aufeis

area ? e.g. a coding of robustness 0 to 3 or a type of error code based on the authors regional and thematic expertise, related to elevation (as the authors describe that too low elevation not as good because early aufeis melt and higher variability, too high not as good because too late snow melt?).

Please also note the supplement to this comment:
https://www.earth-syst-sci-data-discuss.net/essd-2018-99/essd-2018-99-RC1-supplement.pdf
* * *
[Figure]

**Fig. 1.** example visualisation of aufeis data set and attribute names in google earth

**Supplement:**

**Aufeises (naleds) of the North-East of Russia: GIS catalogue for the Indigirka River basin (Russia)**

The GIS database contains the combination of historical data on aufeises (naleds) distribution and their characteristics for the Indigirka River basin from two sources. Historical data origins from the Cadastre of naleds of the North-East of the USSR published in 1958. Other sources are Landsat images for the period 2013-2017.

The database includes:
1. The characteristics of aufeises from the Cadastre of naleds of the North-East of the USSR (1958). The data are stored in file Aufeis_Kadastr.txt

Original version:

| OBJECTID | Index number |
|---|---|
| Auf_Topo_K | Aufeis area (km$^2$) from the Cadastre (1958). If the data was missing, the area was calculated by topographic maps (1980) scale 1: 200 000 |
| kadastr | Index of the aufeis in the Cadastre (1958) (it contains 0 if the aufeis was missing in the Cadastre, but found in the topographic map (1980) scale 1: 200 000) |
| kad_map_nu | Index of the Cadastre (1958) map |
| topo | Presence of the aufeis at topographic map (0 – missing, 1 – present) |
| kad_map | Presence of the aufeis in the Cadastre (0 – missing, 1 – present) |
| toponumber | Nomenclature of the topographic map sheet |
| data | Date of fixing the presence of ice within the aufeis |
| X_geo | Longitude, degree |
| Y_geo | Latitude, degree |
| Altitude | Height above sea level (determined by Aster GDEM), m |
| comment | Comments (mainly typos in the Cadastre map, or the method of determining aufeis area) |
| PolygonID | Index of aufeis by Landsat images (if aufeis is not at the images, the value is missing) |
| Polygon_di | Minimum distance between the aufeis from the Cadastre and the same aufeis from Landsat image (m) |

Suggestions for enhancing the self explanation skills of the attribute names:

| OBJECTID | Index number |
|---|---|
| aufeis data source | Consider to add one general group assignment for this full data set (useful if both data sets are visualized, e.g. in google earth) e.g. aufeis from cadastre |
| aufeis area cadastre sqkm | Aufeis area (km$^2$) from the Cadastre (1958). If the data was missing, the area was calculated by topographic maps (1980) scale 1: 200 000 |
| aufeis index cadastre | Index of the aufeis in the Cadastre (1958) (it contains 0 if the aufeis was missing in the Cadastre, but found in the topographic map (1980) scale 1: 200 000) |
| cadastre index | Index of the Cadastre (1958) map |
| aufeis_in_topo | Presence of the aufeis at topographic map (0 – missing, 1 – present) |
| aufeis_in_map | Presence of the aufeis in the Cadastre (0 – missing, 1 – present) |
| toponumber | Nomenclature of the topographic map sheet |
| date | Date of fixing the presence of ice within the aufeis |
| Long | Longitude, degree |
| Lat | Latitude, degree |
| elevation | Height above sea level (determined by Aster GDEM), m |

| | |
|---|---|
| comment | Comments (mainly typos in the Cadastre map, or the method of determining aufeis area) |
| cross index | cross index of aufeis derived from Landsat (if aufeis is not in Landsat, the value is missing) |
| distance_m | Minimum distance between the aufeis from the Cadastre and the same aufeis from Landsat image (m) |

2. The characteristics of aufeises from Landsat images (2013-2017). The data are stored in file Aufeis_Landsat.txt

Original version:

| | |
|---|---|
| OBJECTID | Index number |
| WRS2ID | The Landsat scene identifier in the WRS2 graph of the US Geological Survey (USGS). The first three digits indicate the column number, the last three digits represent the line number. |
| Date | The date of image |
| Comment | Additional information, for example, if the aufeis was partly covered by clouds and additional images were used to estimate the area |
| ID | Identifier of aufeis by Landsat images (key field for the reference to the Cadastre data=PolygonID) |
| Area_km | Aufeis area by Landsat image, km$^2$ |

Suggestions for enhancing the self explanation skills of the attribute names

| | |
|---|---|
| OBJECTID | Index number |
| aufeis data source | Consider to add one general group assignment for this full data set (useful if both data sets are visualized, e.g. in google earth) e.g. aufeis from Landsat |
| Landsat_WRS2ID | The Landsat scene identifier in the WRS2 graph of the US Geological Survey (USGS). The first three digits indicate the column number, the last three digits represent the line number. |
| Landsat_date | The date of image |
| Comment | Additional information, for example, if the aufeis was partly covered by clouds and additional images were used to estimate the area |
| cross index | Identifier of aufeis by Landsat images (key field for the reference to the Cadastre data=PolygonID) |
| aufeis_area_sqkm | Aufeis area by Landsat image, km$^2$ |
| Add elevation | |

Example of a snapshot visualization of aufeis data and attribute names in google earth

---

## Referee Comment (RC2) · Liljedahl (Referee) · 20 Dec 2018

I much appreciate the effort invested by the authors to not just create a new dataset of recent aufeis coverage, but also in digitizing a historical record and then combining the two to present an interpretation of trends. I concur with the authors, the data will be helpful in refining our understanding of the changing permafrost landscapes.

I had only edits on the English, which I also sent directly to the authors as a word document with track changes. I would also suggest a title like ""Historical and recent aufeis coverage, Indigirka River basin, Russia". I consider the technical aspects solid with no needed changes.

Thank you for making precious Russian research available to the international community!

Anna

Please also note the supplement to this comment:
https://www.earth-syst-sci-data-discuss.net/essd-2018-99/essd-2018-99-RC2-supplement.pdf
* * *
[Figure]

**Supplement:**

**Aufeis of the Indigirka river basin (Russia): the database from historical data and recent Landsat images**

*Olga Makarieva[1,2,], Andrey Shikhov[3], Nataliia Nesterova[2,4], Andrey Ostashov[2]

[1]*Melnikov Permafrost Institute of RAS, Yakutsk*
[2]*St. Petersburg State University, St. Petersburg*
[3]*Perm State University, Perm*
[4]*State Hydrological institute, St. Petersburg*
*RUSSIA*
*omakarieva@gmail.com*

**Abstract:** A dDetailed spatial geodatabase of aufeis (or naled in Russian) within the Indigirka River watershed (, the basin area 305 000 km²), Russia, was compiled from historical Russian publications (year 1958) the Cadaster of aufeis of the North-East of the USSR published in 1958, topographic maps (year xxx), and Landsat images for (year 2013-2017). The aufeis area sharecoverage varies from 0.26 to 1.15% in different river sub-basins within the Indigirka River watershedstudied area. The dDigitized historical archive (Cadaster, (1958) contains the coordinates and characteristics of 897 aufeises with total area of 2064 km². The Landsat-based identification of aufeises for 2013-2017 allowed the description ofincluded 1213 aufeises on with a total area of 1287 km². The combined digital database of the aufeis is available at https://doi.pangaea.de/10.1594/PANGAEA.891036. Accordingly, tThe satellite-derived total aufeis area of aufeis is 1.6 times less than in the Cadaster (1958) dataset. At the same timeHowever, more than 600 aufeis identified by from Landsat images analyses are missing in the Cadaster (1958) archive. It is therefore possible impliesthat the conditions for aufeis formation conditions may have been changed between from the mid-20th century and to the present.

About 60% of total area presents 10% of the largest aufeis. Most aufeis are located in the elevation band of 1 100 – 1 300 m. About 60% of total aufeis area are represented by top presents 10% of the largest of the largest aufeis.

The iInterannual variability of the aufeis area for the period of 2001-2016 was estimated by the example of assessed at the Bolshaya Momskaya naled (aufeis) and the for a group of large aufeis (>xx km2 each) in the basin of the Syuryuktyakh River for the period of 2001-2016. The results of analysis indicate a tendency towards an area decrease in the area of the Bolshaya Momskaya naled in recent years, while no at the same time the reduction Syuryuktyakh Riverin the aufeis area in the basin of the Syuryuktyakh River has not occurredwas observed. The combined digital database of the aufeis is available at https://doi.pangaea.de/10.1594/PANGAEA.891036.

**Keywords:** aufies, the Indigirka river, the Map and Cadaster of aufeises, Landsat images, database, interannual variability, the Bolshaya Momskaya naled (aufeis)

**1. Introduction**

Aufeis (naled in Russian, icings in English) is one of thea glaciation periglacial landforms, standing on the same level with other types o that is characteristic of many streams in cold regions f snow-ice formations and that affecting affects water exchange and economic activity (Alekseev, 1987). They are distributedAufeis are found in permafrost regions, for example, such as Alaska (Slaughter, 1982), Siberia (Alekseev, 1987), Canada (Pollard, 2005), Greenland (Yde and

Knudsen, 2005) and other (Yoshikawa et al., 2007). Aufeis formation
can result in significant economic expenses _as; they_ aufeis may negatively affect
infrastructure and therefore natural resource extraction
(Aufeis of
Siberia…, 1981). Moreover, the springs that often  feed aufeis may in some cases  be the
only source of water  for remote communities (Simakov,
Shilnikovskaya, 1958).
In Russia, aufeis are found in the North-East, Transbaikal region, Yakutia, and West Siberia.
Sokolov (1975)  estimated that the total aufeis water storage in  Russia to be
at least 50 km$^3$, which approximately equals the Indigirka River total annual streamflow.
The main hydrological role of aufeis is the seasonal redistribution of the
groundwater component of river runoff, where the  winter groundwater discharge is released
to summer streamflow through melting of aufeis  (Surface water
resources, 1972). In most cases, the share of the aufeis component in a river's annual streamflow
accounts for 3-7 %, reaching 25-30 % in particular river basins with an extremely large proportion
of aufeis (Reedyk et al., 1995; Kane & Slaughter, 1973; Sokolov, 1975). The most significant
water inflow from aufeis melting takes place in May-June (Sokolov, 1975). For example, the share
of the aufeis flow accounts for more than 11%  of total annual streamflow at the Indigirka
River (gauging station Yurty,  51 100 km$^2$). In May,  aufeis melt may  represent 50
% of monthly total streamflow, but decreases in June to 35 % (Sokolov, 1975).
It is important to understand how climate change may impact aufeis formation
. Aufeis are formed by a complex connection between river and
groundwater. Many studies have report the increase of minimum flow in Arctic
rivers (Rennermalm and Wood, 2010; Tananaev et al., 2016), including those where aufeis are
observed in abundance (Makarieva et al., 2018, in review).  A widely accepted hypothesis for
permafrost regions is that a warming climate  increase the connection between surface
and groundwater  that in turn leads to the increase of streamflow, both
in cold seasons and in annual flow (Bense et al., 2012; Ge et al., 2011; Walvoord et al., 2012;
Walvoord and Kurylyk., 2016). Variation and changes in aufeis extent  can
be assessed using remote sensing  techniques, where aufeis
dynamics can serve as an  indicator of groundwater changes  that are
is otherwise difficult to  observe (Topchiev, 2008; Yoshikawa
et al., 2007).
The understanding of how aufeis respond to a  warming
climate vary. Observations
suggests three to 11 year up and down cycles of aufeis maximum
annual size, s, during which they may vary  up to 25-30 % in comparison with
long-term average values (Alekseev, 2016). However, for the last 50-60 years there is a decrease
in the volume of spring discharge, which feed aufeis
(Alekseev, 2016). Some authors  suggest that degradation
of permafrost in the discontinuous and island-like zones will lead to the decrease of the number of
aufeis and even an almost complete disappearance.  Meanwhile, in the zone of
continuous permafrost in North-East Siberia, a climate warming of 2-3°C
is not projected to  lead to  significant
changes in permafrost extent, but will increase number and size  of both through
and open taliks by the end of the 21th century (Pomortsev et al., 2010).  Such a scenario may
result in dispersion of large aufeis and formation of new small aufeis (Pomortsev et al., 2010).
The projections of increasing dynamics of aufeis formation under climate change are
confirmed by direct observations. In the  valleys of Ulakhan Taryn and Bulus,
central Yakutia, Russia only in four out of 10 aufeis seasons in this century aufeis didn't reach
their maximum area, volume and depth (Pomortsev et al., 2007). In Alaska, however,
no significant changes were documented in

Commented [AL3]: I think you need to list the last name of the author here (not the title) or alternatively, the publisher, and the publication year.

Commented [AL4]: Would be good to include which region this study represents.

Commented [AL5]: Do you mean sporadic permafrost regions?

Commented [AL6]: It is unclear what you are trying to say here.

[revised manuscript text omitted]

[Commented [AL9]: Confusing. Do you mean that there was no recording date provided in the 1958 map, but only in the Cadaster (the catalog)?]

Spatial positioning of the Map of aufeis was conducted using the location description by
Russian topographic maps with the scale of 1:200 000. Grosse and Jones (2011) used the same set
of maps for compiling the dataset of pingos (frost mounds) in northern Asia and described those
maps in details therein. The maps used are based on more detailed maps of 1:50 000 and 1:100
000 scale, which were derived from aerial photography acquired in the 1970–1980's. The use of
1: 200 000 scale guarantees the position assessment precision within 100 m. Each map sheet was
visually searched for aufeis and identified aufeis were marked with an area polygon in a GIS layer.
The locations of 330 aufeis (area 358 km$^2$) were determined based on topographic maps. when
digitized, a point was plotted in the middle of an aufeis at a topographic map.

[Commented [AL10]: Unclear. Do you mean that the 1958 Cadaster/catalog was not solely based on aerial photos, but also through other sources that may not necessarily reflect aufeis coverage in ~1958?]

[Commented [AL11]: Clarify. Did you use these maps or did Grosse use these maps?]

[revised manuscript text omitted]

The head  waters of the Indigirka River, near  the Yurty village, is the  region with the  largest aufeis  coverage (Table 2). Correlation between average elevation of basins and their aufeis coverage (percentage) is statistically significant. Among xx number of basins, the Spearman rank correlation coefficients

**Commented [AL12]:** Please provide some information on how many basins were included.

between the basin average elevation and aufeis percentage are 0.71 and 0.77, the aufeis percentage assessed with the Cadaster and satellite data respectfully.

**4.3 Aufeis area interannual variability**

The assessment of aufeis area interannual variability was conducted in two areas: the Bolshaya Momskaya aufeis , which is located in the Moma River channel ( area in the Cadaster is 82 km$^2$), and for a group of aufeis (total area in the Cadaster is 287.8 km$^2$) in the Syuryuktyakh River basin, which is the left-bank tributary of the Indigirka River. Cloudless images from Landsat-5 (TM), Landsat 7 (ETM+) and Landsat-8 (OLI) were used with the acquisition dates between May 1 and June 30. In the USGS archives, there are no Landsat-5 images for the study territory for the 1984-2007 period. This limits the duration of satellite observations on aufeis to the period since 1999 (when the Landsat-7 satellite was launched). Also, the clouds complicate the acquisition of representative data. The list of the acquisition dates and assessed aufeis area values are presented in Table. 3.

When assessing interannual changes , it is necessary to take the elvation into account. Both areas are located at low elevations (Bolshaya Momskaya  430 to 500 m and Syuryuktyakh 200 to 500 m). which. This contributes to their relatively early and intensive aufeis melt in spring. The aufeis reach their maximum area by the beginning of May. Using the available satellite images it is impossible to make a firm conclusion on aufeis area growth or decline, because the acquisition dates vary significantly from year to year. However, it is possible to make some conclusions based on the available data:

1. In 2002-2017 the Bolshaya Momskaya aufeis did not reach the maximum area stated in the Cadaster (82 km$^2$), even though the satellite image received during the first week of May (2005). when aufeis melting had not yet start. Comparing two images, taken in similar conditions (08.05.2005 and 15.05.2013), it was determined that aufeis area in 2013 was smaller by 18.1 km$^2$ than in 2005.Accordingly, the Bolshaya Momskaya aufeis may have seen a decreasing trend over time in its maximum coverage.

2. The area of the largest aufeis in the Syuryuktyakh River basin in May 2014 was 78.0 km$^2$, which is 8 km$^2$ larger than stated in the Cadaster. It can also be noticed that the maximum aufeis areas in the Syuryuktyakh River basin were detected in the images received at the end of the period (2014-2017), including mid-June (18.06.2015). Therefore, it can be suggested that the aufeis areas within Syuryuktyakh River basin have not decreased since 2002.

**5. Conclusion**

The research conducted here is the first step of the study aimed at the development of a GIS database of the aufeis of North-East Russia. Historical data of the Cadaster (1958) data and topographic maps were used to create a geodatabase of aufeis in the Indigirka River basin (up to the Vorontsovo gauge, with the area of 305 000 km$^2$). It contains historical data on 896 aufeis with total area of 2063.6 km$^2$. Aaufeis detection was conducted for the 2013-2017 period using Landsat imagery with 1213 aufeis identified having a total area of 1287.4 km$^2$ . The historical data from the Cadaster (1958) and satellite data results were combined in the joint Catalogue of aufeis within the Indigirka River basin available at the Pangea repository ( https://doi.pangaea.de/10.1594/PANGAEA.891036).

Recent total aufeis area is 1.6 times smaller than stated in the Cadaster (1958). Simoultaneously, the historical Cadaster archive is lacking data on over 600 aufeis that were identified using satellite images. This suggests  that the Cadaster data is incomplete, while there may also have been significant change in aufeis formation conditions in the last half century. One of the

**Commented [AL13]:** Can you make a stronger conclusion? For example that the total aufeis area have decreased over time, while simultaneously it appears that additional aufeis may have formed over time.

[revised manuscript text omitted]

---

## Author Comment (AC1) · 5 Feb 2019

Reviewer 1 This is a unique temporal data collection of aufeis data in the Indigirka river basin, Russia. Aufeis or naleds deposits are thick accumulations of ice that form during winter along stream and river valleys in arctic and subarctic regions impacting hydrology and geomorphology of these regions. The authors compiled and standardized historical data on aufeis deposits in the Eastern Siberian Indigirka river basin from a historical Russian National cadastre complementing data using historical topographical maps and added a new data set on aufeis derived from Normalized Differential Snow Index (NDSI) index calculation using Landsat 8 OLI sensor data. The authors cross-referenced the historical and the present-day data collection. The data collection is organised as a Geographic Information System GIS data base including data on location, area coverage, elevation, time stamp, source of data in form of attribute tables and the aufeis objects in the data format of GIS point and polygonal vector layers. The Indigirka aufeis catalogue is published on PANGAEA in the form of a GIS data base with a helpful and detailed read-me description of the attribute tables. The data collection will be of interest to hydrologists, climatologists, geomorphologists, cryologists and social science. The authors document in the manuscript the generation of the historical and the modern date data sets and its meta data characteristics. The authors also discuss the validity of data, the cross referencing between historical and nowadays aufeis deposits and reasons for mismatches in areal coverage and locations and possible changes due to climate.

Comment: The paper is in general clearly written with many details provided. However, the article including the title, the PANGAEA data publication including title, abstract and the metadata description need to be carefully edited for English before acceptance of the paper. The data compilation process and metadata is not thoroughly and clear enough shown and explained and the GIS data require further standardization and optimization to make them reusable.

Technical issues, GIS data: 1) the GIS shape files contain different projections: The GIS data catalogue is published in PANGAEA as an ARCGIS project data base. The downloaded data base is user friendly readible and usable using the proprietary GIS software ARCGIS. ARCGIS licenses are costly and many user groups may use open source GIS or other geodata software packages. Using ARCGIS software the shape-files are automatically but only virtually brought to the same projection. The GIS shape files are also readable and reusable using open source geodata software packages – however the 2 data collections have different projections (the aufeis kadastr shape file contains the projection "Asia_North_Lambert_Conformal_Conic" the aufeis Land-sat shape file not). This requires users of these datasets who are using free software packages to reproject the shape files to a common projection prior to being able to use the data sets together. Please standardise the shapefiles using one projection

Response: We prepared the data according to the comments. The GIS database contains the data of aufeis in two forms: ArcGIS 10.1/10.2 and Qgis 3* projects. All data and projects have WGS 1984 coordinate system (without projection). ArcGIS and Qgis projects contain two layers, such as Aufeis_kadastr (historical aufeis data collection, point objects) and Aufeis_Landsat (satellite-derived aufeis data collection, polygon objects).

Comment: 2) the GIS attribute files do not contain self-explanatory attribute names: The Indigirka aufeis data collection is a highly valuable data set, specifically also because the authors are using cross reference indices to link the data sets. This needs to be made more clear in the naming and cross-referencing of the attribute names. E.g., the cross reference index should be also named accordingly, e.g. as cross index similarly in both attribute tables, not named ID in the aufeis_Landsat data set and named PolygonID in the aufeis kadastr data set. Naming of similar attributes should be standardized between the data sets, e.g. the attribute area in sqkm. Suggestions on attribute naming is attached as supplement. Please consider to change attribute names to more self-explanatory names.

Response: We followed the suggestions on enhancing attribute naming as much as possible. Though due to the limited length of the name we could not do it in all namings. See the Tables 1 and 2 in the paper. The PANGAEA database is updated accordingly.

Comment: The data set can also be uploaded in Google Earth with visualization of the data objects and the metadata and will be by this very easy re-usable if attribute naming and cross-referencing between the 2 data sets will be made as self-explanatory as possible.

Response: We uploaded the database into Google Earth and added the files to PANGAEA database. Additionally the watershed borders which are mentioned in the analysis in the paper added in Google earth format.

Comment: 3) consistency of published GIS data with manuscript content: Authors show in the manuscript assessments of both data sets – cadastre derived and satellite derived related to elevation. The attribute elevation is however missing in the attribute table of aufeis_Landsat. Consider to add information on elevation into the attribute table of the aufeis Landsat data set.

Response: The attribute Elevation is added to Landsat data set (See also Table 2 in the paper).

Comment: Issues, data publication on PANGAEA: Title: aufeis is the plural form of aufeis, the plural form aufeises does not exist.

Response: We changed the title of the database to "Aufeis (naleds) of the North-East of Russia: GIS catalogue for the Indigirka River basin (Russia)"

Comment: Abstract: The abstract should be extended to contain more technical information on the data. Authors should inform the users that the data download will consist of a complete ARCGIS project containing 2 different feature GIS shape files with historical and the nowadays aufeis data collection. The authors can add short information in the abstract on how the data were generated. Very useful for future users of the GIS data is to provide in the abstract text information on the projection of the GIS data collection – this is sometimes handy for reading data in in some open source geodata software packages.

Response: We extended the abstract as the following. The GIS database contains the data of aufeis (naleds) in the Indigirka River basin (Russia) from historical and nowadays sources, and complete ArcGIS 10.1/10.2 and Qgis 3* projects to view and analyze the data. All data and projects have WGS 1984 coordinate system (without projection). ArcGIS and Qgis projects contain two layers, such as Aufeis_kadastr (historical aufeis data collection, point objects) and Aufeis_Landsat (satellite-derived aufeis data collection, polygon objects). Historical data collection is created based on the Cadastre of aufeis (naleds) of the North-East of the USSR (1958). Each aufeis was digitized as point feature by the inventory map (scale 1:2 000 000), or by topographic maps. Attributive data was obtained from the Cadastre of aufeis. According to the historical data, there were 896 aufeis with a total area 2063.6 km2 within the studied basin. Present-day aufeis dataset was created by Landsat-8 OLI images for the period 2013-2017. Each aufeis was delineated by satellite images as polygon. Cloud-free Landsat images are obtained immediately after snowmelt season (e.g. between May, 15 and June, 18), to detect the highest possible number of aufeis. Critical values of Normalized Difference Snow Index (NDSI) were used for semi-automated aufeis detection. However, a detailed expert-based verification was performed after automated procedure, to distinguish snow-covered areas from aufeis and cross-reference historical and satellite-based data collections. According to Landsat data, the number of aufeis reaches 1213, with their total area about 1287 km2. The difference between the Cadastre (1958) and the satellite-derived data may indicate significant changes of aufeis formation environments.

Comment: The authors could add an overview figure of the data set as additional information.

Response: We uploaded the database into Google Earth and added the files to PANGAEA database. Additionally the watershed borders which are mentioned in the analysis in the paper added in Google earth format. We also added overview figure to the database (Fig 1).

Comment: Published data: the authors published the GIS project with 2 feature layer data and the 2 data collections also in form of ASCII files and a detailed read me word file documenting the attribute tables. Information on the GIS project itself in the read-me file is missing: e.g., information on the format (ARCGIS) and projection.

Response: We added the missing information.

Comment:- Issues, manuscript: General: aufeis is the plural form of aufeis, the plural form aufeises does not exist. Authors could also consider to sometimes refer to aufeis deposits in the manuscript if this fits.

Response: We fixed wrong plural form through the text and figures.

Comment: Authors could refer to the cadastral map instead of map throughout the text, also to better distinguish for the reader the cadastral map from topographic map forms.

Response: The expression "Cadastral map" has been introduced starting from Line 133 after the description of the Cadsatre.

Comment: Abstract: The authors should enrich the abstract with much more information on the technical generation and technical contents of the data set and with less discussion on changed areas and potential reasons that would be kind out of scope and not the focus of this ESSD publication. A great meta data information in this data collection is the cross-reference index enabling users of this data set to link and compare these very different 2 data set types: the historical and the nowadays aufeis data sets.

Response: Short information on Landsat-based aufeis detection and cross-reference index is added in the abstract. Lines 13-16: Identification of aufeis by late-spring Landsat images was performed with a semi-automated approach according to Normalized Difference Snow Index (NDSI) and additional data. Then, a cross-reference index was set for each aufeis, to link and compare historical and satellite-based aufeis data sets.

Comment: keywords: reconsider the keywords, e.g., aufeis, Indigirka, Bolshaya Momskaya, Land-sat, NDSI, cadastre, cadastral map;

Response: We changed the keywords according to the comment. Line 34-35. Keywords: aufeis, Indigirka, Landsat, NDSI, Cadastre, Cadastral map, Bolshaya Momskaya aufeis

Comment: Introduction: authors should provide an explanation what is aufeis in the first sentences of the introduction. That aufeis are thick accumulations of ice that form during winter along stream and river valleys in arctic and subarctic regions.

Response: We provided the explanation. Lines 38-40. Aufeis (naleds in Russian, icings in English) are the accumulations of ice that are formed by freezing underground, surface and atmospheric waters on the surface of the earth or ice along streams and river valleys in arctic and subarctic regions.

Comment: 2 Research objective: this subtitle is misleading as the motivation of this study and data set compilation is already well introduced by the authors in the introduction chapter. This chapter describes the study region. Please add an overview figure with the geographical setting of the Indigirka river basin and the extent of the data set in relation to Eastern Siberia. E.g., Figure 6 is already to zoomed in to provide this information.

Response: We changed this subtitle to Study region (line 105).An overview figure with the geographical location of the Indigirka river basin is added (Line 484). Fig. 2 Geographical location of the Indigirka river basin

Comment: 3 Material and Methods: The authors should add the tables from the published read me file in the respective subsections 3.1 and 3.2.

Response: Table 1 and 2, which contain the structure of the GIS database of aufeis according to Cadastre and Landsat images has been added.

Comment: The authors should add flow chars to make their data processing steps more clear in the in the respective subsections 3.1 and 3.2. For example the role of the thalweg creation remains unclear to the reader.

Response: Thalweg creation was an essential step of semi-automated separation of the aufeis from snow-covered areas by late-spring Landsat images. Indeed, almost all aufeis are located either at streams or thalwegs, or in immediate proximity to them. On the contrary, the snow cover in late spring mainly remains on mountains ridges and other areas with high altitude, e.g. relatively far from thalwegs. Based on the preliminary analysis of aufeis location in relation to created network of thalwegs, we estimated, that 1.5 km wide buffer zone around the thalwegs covers almost all aufeis. So, snow and ice covered areas, which are located outside this buffer, are excluded from the further analysis. The explanation has been added. Line 213-223: Aufeis detection algorithm was realized in ArcGIS with the help of the ModelBuilder application. Apart from the Landsat images, the digital terrain model (DTM) GMTED2010 (Danielson and Gesch, 2011) with the spatial resolution of 250 m was used to build a network of thalwegs within the study basin. This is essential for semi-automated separation of the aufeis from snow-covered areas by late-spring Landsat images. Indeed, almost all aufeis are located either at streams or thalwegs, or in immediate proximity to them. On the contrary, the snow cover in late spring mainly remains on mountains ridges and other elevated locations, e.g. relatively far from thalwegs. Based on the preliminary analysis of aufeis location in relation to created network of thalwegs, we found, that 1.5 km wide buffer zone around the thalwegs covers almost all aufeis. So, snow and ice covered areas, which are located outside this buffer, are excluded from the further analysis.

Comment: The ASTER GDEM data set needs to be introduced and explained as the meta data information on elevation is taken from this digital data set. Also for the Landsat derived dataset? This does not become clear to the reader.

Response: We added the information about DEM. Line 214-216: Apart from the Landsat images, the digital terrain model (DTM) GMTED2010 (Danielson and Gesch, 2011) with the spatial resolution of 250 m was used to build a network of thalwegs within the study basin.

Comment: 3.2. The level of the USGS Landsat data product that was used remains unclear. The authors did not use the Landsat T1 Level2 (L2) that is the surface reflection coefficient already? Did the authors use the Landsat T1 Level1 data products that are terrain-corrected (T1) and Top-of-Atmosphere radiances (L1)? Because authors refer to brightness?

Response: We used Landsat 8 collection 1 Level1T (terrain-corrected) data products. The explanation has been added. Line 204-205: We used Landsat 8 collection 1 level-one terrain-corrected product (L1T) with radiometric and geometric corrections

Comment: The authors describe: Preprocessing of the images (transformation brightness into reflection coefficient) was performed with the use of Semi-Automatic Classification Plugin module in QGIS 2.18. Does it mean that an atmospheric correction was performed to surface reflection coefficient? Which type of atmospheric correction was performed to come to the surface reflection coefficient / surface reflectance?

Response: Preprocessing of the images was performed with the use of Semi-Automatic Classification Plugin module (QGIS 2.18). It includes the calculation of surface reflectance and atmospheric correction by Dark Object Subtraction (DOS1) image-based algorithm, described by (Chavez, 1996). The explanation has been added. Line 209-212: Preprocessing of the images was performed with the use of Semi-Automatic Classification Plugin module (QGIS 2.18). It includes the calculation of surface reflectance and atmospheric correction by Dark Object Subtraction (DOS1) image-based algorithm, described by (Chavez, 1996).

Comment: 3.3 A good description of the cross reference between the aufeis deposits in the historical aufeis data collection and the nowadays data collection is missing. Authors can consider to add a short sub-paragraph 3.3. It would be helpful for re-using the data set if authors put some details here, e.g. highlight that there is the cross reference ID in both attribute tables.

Response: The sub-paragraph is added Line 256-271: Đąross-verification of aufeis data collections by the Cadastre (1958) and satellite imagery was performed in two steps. At the first step, we found closest aufeis in the Landsat-derived dataset for each aufeis from the Cadastre data, if the distance between them was less than 5000 m. The determination of search radius is based on a preliminary analysis of the aufeis locations by the Cadastre in relation to Landsat-based dataset. As a result, the cross index (identifier of the closest aufeis in the Landsat-derived dataset) and minimum distance (m) to the closest aufeis were determined for aufeis from Cadastre. For Landsat-based dataset, the cross index is the key field for the reference to the dataset from Cadastre. At the second step, a full manual verification was performed to found the mistakenly interrelated aufeis. For example, if the closest aufeis from Cadastre and from Landsat-based dataset were at a distance of less than 5000 m, but in different thalwegs, they were considered as different (unrelated) aufeis. In total, 260 aufeis from Cadastre were not verified by Landsat images. For them, the NoData value (–9999) was set in the Cross Index and Distance fields of attributive table (see Table 1 with the structure of GIS dataset from Cadastre).

Comment: 4 Results and verification: The chapter does not seem to describe or focus on verification?

Response: We changed to subtitle "Results" (line 273)

Comment: In the first section of 4 Results the authors very interestingly assess the linkages and differences between the data sets – this could become a subchapter 4.1. with a title relating to the comparison of the historical to the modern data collection. All of the above points can be addressed with minor corrections, just a few sentences or less.

Response: We corrected the title to "Comparison of the historical and modern data collection "(line 274)

Comment: consider adding a Discussion chapter with a short discussion about the usability of this data set on aufeis area growth or decline, could be one outcome of your study on the variability to assign higher variability and lower accuracy to the extraction of the aufeis area at lower elevation? Would it be possible to assign different reliability (consistency of measurement) levels for the representativeness of the derived aufeis area ? e.g. a coding of robustness 0 to 3 or a type of error code based on the authors regional and thematic expertise, related to elevation (as the authors describe that too low elevation not as good because early aufeis melt and higher variability, too high not as good because too late snow melt?).

Response: We added the Discussion section. We do not think we may assign relative reliability; instead some general analysis of the data limitations (lines 367-421) is presented.

Please also note the supplement to this comment:
https://www.earth-syst-sci-data-discuss.net/essd-2018-99/essd-2018-99-AC1-supplement.pdf
* * *
**Fig. 1.** Google Earth aufeis database overview

60˚ E    40˚    20˚    0˚    20˚    40˚    60˚

RUSSIA

60˚
N

CANADA

USA

500   1 000            2 000 km

Indigirka river basin

[revised manuscript text omitted]

---

## Author Comment (AC2) · 5 Feb 2019

The comments were sent in the form of doc document. So we have combined them here.

Comment: Suggestion to change the title to "Historical and recent aufeis, Indigirka River basin, Russia"

Response: Accepted. We changed the title.

Comment: In the abstract to specify present or historical aufeis are located in the elevation band of 1000-1300 m.

Response: Specified. Line 25: Most present and historical aufeis are located in the elevation band of 1000 – 1200 m.

Comment: Suggestion to the reference. I think you need to list the last name of the author here (not the title) or alternatively, the publisher, and the publication year.

Response: We changed the reference as the following: (Aufeis of Siberia..., Nauka, 1981) – Line 46

Comment: The question to the reference (Alekseev, 2016) - Would be good to include which region this study represents.

Response: Line 74-77 Expanded the sentence as the following: However, the same author (Alekseev, 2016) states a general tendency to the decrease of aufeis volume for the last 50-60 years in some aufeis-affected areas of Russia such as Baikal region, South Yakutia, Kolyma region, Eastern Sayan Mountains, following the increase of global and local air temperature.

Comment: Unclear. Is it 896 or 808 aufeis in your database?

Response: Lines 156-157. Clarified. Our compilation contains data on 896 aufeis. The aufeis are presented as point objects in our database. The areas are specified only for 808 aufeis.

Comment: Confusing. Do you mean that there was no recording date provided in the 1958 map, but only in the Cadastre (the catalog)? Do you mean that the 1958 Cadastre/catalog was not solely based on aerial photos, but also through other sources that may not necessarily reflect aufeis coverage in ∼1958?

Response: Lines 162-166. Clarified. The dates of ice recording for the remaining 34 % of the aufeis were not described, meaning that aufeis detection could be carried out based not on the visible ice presence at the aerial images but on geomorphological features of river valleys. Therefore, the Cadastre might as well contain data on old aufeis glades, where the aufeis themselves were absent.

Comment: Clarify. Did you use these maps or did Grosse use these maps?

Response: Lines 167-172. Clarified. Spatial positioning of the Cadastral Map of aufeis was conducted using the location description by Russian topographic maps with the scale of 1:200 000. Grosse and Jones (2011) used the same set of maps for compiling the dataset of pingos (frost mounds) in northern Asia and described those maps in details therein. The maps of 1:200 000 scale were based on more detailed maps of 1:50 000 and 1:100 000 scale, which were derived from aerial photography acquired in the 1970–1980's.

Comment: Please provide some information on how many basins were included in correlation analysis.

Response: Lines 328-330. Clarified. Among 6 basins, the Spearman rank correlation coefficients between the basin average elevation and aufeis percentage are 0.71 and 0.77, the aufeis percentage assessed with the Cadastre and satellite data respectfully.

Comment: Can you make a stronger conclusion? For example that the total aufeis area have decreased over time, while simultaneously it appears that additional aufeis may have formed over time.

Response: We added the discussion section (lines 367-421) which shows the limitations of the datasets. Also we added the analysis of area reduction of large and giant aufeis. We do not think we have the complete evidence for strong conclusion. In Conclusion section we are rather cautious: Lines 436-439: The analysis of large and giant aufeis seems to indicate that there has been a significant decrease in aufeis area over the period of last 70 years. Additional analysis of historical aerial photography data could help to clarify the issue of aufeis area decline trend since the middle of the 20th century to the present.

Comment: Spelling of aufeis through the text and in the figures

Response: Corrected in the text and figures.

Please also note the supplement to this comment:
https://www.earth-syst-sci-data-discuss.net/essd-2018-99/essd-2018-99-AC2-
supplement.pdf

---

## Author Comment (AC3) · 5 Feb 2019

Comment: This is a fascinating analysis and I congratulate the authors on an excellent paper. I am very interested in where they might take this analysis from here. It seems to me that based on their data, there appears to be a declining trend in the number and/or size of aufeis in the study area. I am wondering if the authors could access additional satellite and/or aerial photography or other historical records that might allow them to determine if such trends are in fact occurring and attempting to quantify them.

Response: We added the Discussion section where describe the limitations of both datasets and provide the analysis of the area reduction of large and giant aufeis. The main limitation of the historical aufeis dataset is that the Cadastre provides an area of aufeis glades, but not the aufeis themselves. The satellite-derived assessment of the aufeis area has the other source of uncertainty. It is often impossible to determine the maximum area of aufeis by satellite images, since it is observed at the beginning of snow melt season, when aufeis are still covered with snow. In late spring and beginning of summer, the area of aufeis may already been significantly reduced in comparison with the maximum values, due to melting and mechanical destruction. The analysis of large and giant aufeis seems to indicate that there has been a significant decrease in aufeis area over the period of last 70 years. Additional analysis of historical aerial photography data could help to clarify the issue of aufeis area decline trend since the middle of the 20th century to the present. We plan to conduct such analysis in the future.

Please also note the supplement to this comment:
https://www.earth-syst-sci-data-discuss.net/essd-2018-99/essd-2018-99-AC3-supplement.pdf

**Supplement:**

[revised manuscript text omitted]